# The Clinical Benefit of Percutaneous Transhepatic Biliary Drainage for Malignant Biliary Tract Obstruction

**DOI:** 10.3390/cancers14194673

**Published:** 2022-09-26

**Authors:** Ivan Nikolić, Jelena Radić, Andrej Petreš, Aleksandar Djurić, Mladjan Protić, Jelena Litavski, Maja Popović, Ivana Kolarov-Bjelobrk, Saša Dragin, Lazar Popović

**Affiliations:** 1Oncology Institute of Vojvodina, Department of Medical Oncology, Put Doktora Goldmana 4, 21204 Sremska Kamenica, Serbia; 2Faculty of Medicine Novi Sad, University of Novi Sad, Hajduk Veljkova 3, 21000 Novi Sad, Serbia; 3Clinical Center of Vojvodina, Department of Radiology, Hajduk Veljkova 1-9, 21000 Novi Sad, Serbia; 4Department for Animal Sciences, Faculty of Agriculture, University of Novi Sad, 21000 Novi Sad, Serbia

**Keywords:** biliary drainage, jaundice, malignant obstruction, adequate bilirubin decline (ABD)

## Abstract

**Simple Summary:**

Percutaneous transhepatic biliary drainage (PTBD) has been widely employed as a biliary decompression procedure for malignant proximal biliary obstruction. Patients underwent PTBD procedures of jaundice caused by malignant disease for restarting chemotherapy or palliative intent. Clinical outcomes following PTBD procedure in the two groups of patients, according to the adequate bilirubin decline (ABD) needed to restart chemotherapy, are presented in this analysis. Median survival time following the PTBD was 9 weeks, but in patients with ABD who received chemotherapy it was 64 weeks. Patients with the longest survival rate were in good performance status (ECOG 0–1) and lower bilirubin (<120 µmol/L) and LDH (<300 µmol/L) levels at the time of the procedures. Improving quality of life is a major goal in this palliative treatment, but we really need to assess the potential benefits and risks and determine how to select patients who would have positive outcome for this procedure.

**Abstract:**

Percutaneous transhepatic biliary drainage (PTBD) is a decompression procedure for malignant proximal biliary obstruction. In this research, over a six-year period, 89 patients underwent PTBD procedure for jaundice caused by malignant disease to restart chemotherapy or for palliative intent. Clinical outcomes after PTBD procedure in the two groups of patients, according to the adequate bilirubin decline (ABD) needed for subsequent chemotherapy, are presented in this paper. Survival and logistic regression were plotted and compared using Kaplan–Meier survival multivariate analysis with a long-range test. Results were processed by MEDCALC software. In the series, 58.4% (52/89) of patients were in good performance status (ECOG 0/1), and PTBD was performed with the intention to (re)start chemotherapy. The normalization of the bilirubin level was seen in 23.0% (12/52), but only 15.4% (8/52) received chemotherapy. The median survival time after PTBD was 9 weeks. In patients with ABD that received chemotherapy, the median survival time was 64 weeks, with 30-day mortality of 27.7%, and 6.4% of death within 7 days. The best outcome was in patients with good performance status (ECOG 0–1), low bilirubin (<120 µmol/L) and LDH (<300 µmol/L) levels and elevated leukocytes at the time of the procedures. PTBD is considered in ABD patients who are candidates for chemotherapy.

## 1. Introduction

Percutaneous transhepatic biliary drainage (PTBD) has been widely employed as a biliary decompression procedure for malignant biliary obstruction [1]. Tumor-induced obstructive jaundice can be caused by Klatskin tumors (hilar cholangiocarcinoma), pancreatic adenocarcinoma, gallbladder carcinoma, metastases in the porta hepatis lymphnodes, distal cholangiocarcinoma, or hepatocellular carcinoma (HCC) [2,3]. In the majority of patients, malignant obstructive jaundice is incurable with a bad prognosis [4]. Different modalities are available for biliary drainage, such as surgical drainage, endoscopic retrograde cholangiopancreatography (ERCP) or percutaneous transhepatic biliary drainage (PTBD). None of these procedures have been proven superior in comparison to others, and the question of the most effective procedure for the decompression of bile duct obstruction remains controversial [5]. PTBD facilitates external and internal biliary tree drainage and is the primary method of relieving biliary obstruction from malignant lesions above the level of the common hepatic duct (proximal obstruction). Patients with distal biliary obstructions (MDBO) are treated by ERCP or endoscopic ultrasound biliary drainage (EUS-BD) [6]. A recently published meta-analysis compared the efficacy and safety of PTBD and ERCP in resectable peri-hilar cholangiocarcinoma patients, and PTBD resulted in less conversion and lower rates of pancreatitis and cholangitis [7]. In addition, PTBD has been shown to bear the risk of seeding metastases [8]. The major goal of palliative treatment of obstructing biliary tumors is the restoration of bile flow to the intestine [9]. Median survival has been found to be significantly longer in patients with restored bile drainage (4.8–11.8 months) regardless of technique, than in patients with failed attempts to restore biliary drainage (1.3–1.8 months) [10,11]. Biliary obstruction may impair quality of life due to pruritus, cholangitis and liver failure [12]. In patients who are unsuitable for curative resection, the relief of obstructive jaundice improves quality of life [13]. Furthermore, the role of biliary drainage prior to subsequent treatment remains controversial. Biliary drainage aims to lower the hyperbilirubinemia, in order to either relieve jaundice-related symptoms, or to enable the administration of systemic therapy [14]. In accordance with this, the potential beneficial effects of chemotherapy on patient survival following PTBD have been shown in several studies [15,16]. However, biliary drainage carries the risk of adverse events, such as bleeding, pain, bile leak, cholangitis, and pancreatitis or stent dysfunction. Clinical outcomes following PTBD procedure that depend on adequate bilirubin decline (ABD) required for subsequent chemotherapy are presented in this analysis. These events may negatively impact survival and the quality of the last phase of the patient’s life [17]. Reduction of bilirubin levels following biliary drainage and relief of obstruction were found to indicate a gradual recovery of hepatic function [18].

The main focus of the presented study is to define clinical and laboratory parameters in patients with malignant biliary tract obstruction that could be used as a tool to select the best candidates for PTBD procedure.

## 2. Materials and Methods

This was a retrospective analysis of prospectively maintained data over six years, from January 2016 to December 2021, at the Department of Digestive Oncology, Clinic for Medical Oncology, Oncology Institute of Vojvodina in Sremska Kamenica, Serbia. During this period, a total of 89 patients with proximal biliary obstruction underwent PTBD procedures because of jaundice caused by malignant metastatic (stage IV) disease for pre-operative or palliative intent. According to the standard protocol inthe Oncology Institute, PTBD is the primary method of relieving biliary obstruction from malignant lesions above the level of the common hepatic duct (proximal biliary obstruction), as opposed to ERCP, which is the primary method for relieving distal biliary obstruction. Since this is a group of patients in which systemic treatment is contraindicated due to elevated bilirubin levels, it is not possible to choose a control group with systemic treatment only. On the other hand, the goal of the procedure is to enable systemic therapy after the adequate bilirubin level decline, so it was not possible to form a group that remained on follow-uponly after the endoscopic procedure. The aim of this study was to show what percentage of patients with malignant obstructions have a potential benefit from these procedures, and we used as a control the group of patients without adequate bilirubin decline and thus without oncologic treatment. The indication for bile duct re-canalization was either intention to restart chemotherapy, cholangitis or palliative symptom relief. Patients or their families provided written informed consent. The patients’ demographics, Eastern Cooperative Oncology Group (ECOG) performance status, clinical and laboratory parameters, serum bilirubin levels prior to and following the drainage, primary cause of obstruction, additional treatment and survival were analyzed. Complications associated with the intervention were analyzed. Electronic clinical records (BIRPIS) of all patients were reviewed. We noted the type of primary tumor and classified it: colorectal cancer, pancreas tumor, cholangiocarcinoma, gastric and the other tumors (miscellaneous group). We followed the timeline from diagnosis of disease to biliary duct obstruction and data about chemotherapy treatment before and after PTBD.

Clinical success was defined in some studies by decreases in serum bilirubin levels of >20% within 7 days after drainage, compared to the bilirubin level prior to the procedure [19]. Patients were usually discharged from the hospital when they recovered from peri-procedural discomfort and morbidity, and serum bilirubin showed a downward trend. A 6-week period is an acceptable time frame for the satisfactory return of liver function, based on literature data and on the previous experience [17]. PTBD-related mortality was registered within 30 days of the procedure or intrahospitally. Predictors of 30-day mortality were the following: type of tumor (colorectal versus other tumors, performance status (ECOG 0, 1 vs. 2 and 3), and bilirubin higher than 300 µmol/L [20]. This cut-off level was identified as an independent predictor for early complications (1–30 day) by the study of Tapping et al. [21]. In our analysis, the cut-off level of bilirubin was 120 µmol/L.

Survival was calculated by the number of days since PTBD until mortality or until the end of the study (December 2021).

Statistical analysis was performed using MEDCALC Software. Kaplan–Meier survival analysis with long-range test was used to plot and compare survival, and logistic regression analysis was used for multivariate analysis.

## 3. Results

Among the 89 patients included in this analysis, 61.8% (55 patients) were male. The mean age of the patients at the time of the procedures was 62.82 years (range 40–84 years).

A total of 32.6% of patients (29 patients) had diagnosis of colorectal cancer. Metastatic pancreatic adenocarcinoma, cholangiocarcinoma, breast cancer, ovarian cancer and gastric cancer were other causes of the obstructive jaundice. In total, 97 procedures were performed on 89 patients with metastatic disease. Seven patients underwent another PTBD procedure for recurrent biliary obstruction with a time interval of 18 months. The characteristics of the patients are summarized in Table 1.

The median level of bilirubin before the procedure was 290 µmol/L (with a range of 70–623 µmol/L). In our analysis, PTBD was considered to be successful in the case of the normalization of the bilirubin level with the possibility of re-starting chemotherapy. The median survival rate of patients with normalization of the bilirubin level was 32 weeks versus 8 weeks in patient’s without bilirubin level improvement, *p* < 0.0001 HR 0.27 (0.16–0.45) (Figure 1).

At the time of PTBD, the cut-off level of bilirubin of 120 µmol/L differentiates patients according to their survival rate. With this cut-off, there is a significant difference in the median overall survival rate between patients with lower and higher levels of bilirubin (31 vs. 8 weeks) *p* = 0.0013, HR 0.35 (0.18–0.66) (Figure 2).

Only 15.7% (14/89) of patients achieved a normalized bilirubin level, and after that, nine of them (9/14) received chemotherapy; these patients had amuch longer median survival rate—64 weeks versus 9 weeks with *p* < 0.0001 HR 0.27 (0.16–0.48) (Figure 3).

Depending on the ECOG status, we divided the patients in two groups: the ones in good condition (ECOG 0–1) 58.4% (52/89) who were candidates for additional chemotherapy after PTBD, and the other group with poor performance status (ECOG 2–3) 41.6% (37/89) who were not candidates for chemotherapy. The median survival rate of patients in good performance status (ECOG 0–1) was 11 weeks versus 8 weeks in poor-condition patients *p* = 0.0010 HR 0.41 (0.24–0.69) (Figure 4). 

In the group of patients with good ECOG, we registered the normalization of bilirubin levels in 23% (12/52), and 8 (8/12) patients received additional chemotherapy. Patients without a significant difference in the median survival rate between groups were mostly those with colorectal cancer: colorectal versus non-colorectal cancer patients (9 weeks versus 10 weeks) *p* = 0.61 HR 1.12 (Figure 5). 

In a group of patients with poor ECOG, only one patient (1/2) with pancreatic cancer received chemotherapy (Table 2).

In the group of patients where additional chemotherapy was not the primary goal, because of the poor general condition of patients, PTBD was performed to relieve the symptoms of jaundice (pruritus, pain, cholangitis and liver failure) and to improve their quality of life.

Before malignant biliary obstruction, only 38.2% (34/89) of patients had already received chemotherapy and half of them (17/34) with more than one line of chemotherapy. 

The median survival time for all patients was 9 weeks or 63 days with a 7-day mortality of 6.4% and a 30-day mortality of 27.7% (Figure 6).

Among the laboratory parameters, a significant difference in overall survival could be predicted based on the number of leukocytes (11 vs. 4 weeks) *p* = 0.009 HR 0.48 (0.27–0.83) (Figure 7) and if LDH was <300 µmol/L (10 vs. 4 weeks) *p* = 0.0022, HR 0.34 (0.17–0.68) (Figure 8).

In a multivariate analysis, LDH, leukocytes, adequate bilirubin decline, ECOG, and following chemotherapy were parameters that were found to be predictive of longer survival rate in patients with adequate bilirubin decline (Table 3).

As for other laboratory parameters (hemoglobin < 120 µmol/L, level of platelets, aspartate aminotransferase (AST), alanine aminotransferase (ALT), creatinine, alkaline phosphatase < 500 µmol/L), we did not find significant difference in the survival rate of patients. In addition, patient age (<60 years), primary tumor type and number of previous lines of chemotherapy were without influence on the survival rate in our analysis. 

Complications were reported in 27% of procedures (Table 4). Among them, pain and biliary leak were most frequent. Bleeding (defined as a drop of hemoglobin level by more than 20 g/L and need for transfusion) occurred in 5% of the procedures. 

## 4. Discussion

Malignant biliary obstruction is often caused by external compression from lymph node metastases or internal stricture from neoplasm. Obstructive jaundice caused by metastatic disease of advanced carcinoma leads to a rapid deterioration of the performance status and a short survival. Percutaneous biliary drainage and stenting are established, and well-reported methods are used to relieve jaundice. Rates of clinical success of biliary drainage vary widely between the reported studies, from 42 to 81% [20,22,23,24], but clinical success is defined in some studies as a decrease in serum bilirubin levels by 20% within 7 days after drainage [25]. In our analysis, the main goal was the normalization of the bilirubin level with additional oncologic treatment. All procedures were technically successful, and 15.7% of procedures were biochemically successful with a resolution of jaundice and normalization of the bilirubin level. If we analyze only patients where additional chemotherapy after PTBD was the goal, that percentage increases to 23%. In our analysis, we had 7.9% (7 patients) patients who underwent additional PTBD. Improving quality of life is a major goal in the palliative treatment procedure for recurrent biliary obstruction. A 2008 review of PTBD series reported the recurrence of obstructive jaundice in 5–25% of patients, with the majority undergoing a repeat PTBD [8].Surgical or endoscopic drainage of the biliary system was not examined in this study. Surgical drainage is associated with a higher postoperative mortality and morbidity and an increased length of hospital stay compared to nonsurgical intervention [26]. The choice between endoscopic or percutaneous biliary drainage is less clear, as few randomized trials exist [27]. The decision often depends on the level of biliary obstruction. PTBD is recommended for lesions above the common hepatic duct, as opposed to endoscopic drainage, which is the recommended procedure for relieving distal bile duct obstruction. 

PTBD can be performed for symptom control and treatment of cholangitis; in patients with a good performance status, it allows for palliative chemotherapy. In our series, 58.4% (52/89) patients were in the good performance status group (ECOG 0,1) and PTBD was performed with the intention to re-start chemotherapy. A 6-week period is an acceptable time frame for satisfactory return of liver function, based on the literature data and on previous experience [17]. In this group of patients, normalization of the bilirubin level was seen in 23.0% (12/52) of patients but only 15.4% (8/52) received chemotherapy, mostly patients (6/8) with metastatic colorectal cancer. This rate is lower if compared with the reported series of Vandenabeele L. [20] and Meller M.T. [28] with 34% and 22%.

High serum bilirubin levels often represent contraindications for surgical chemotherapy, radiotherapy and local methods of treatment, and those patients should only receive supportive care. Since more than 80% of our patients did not reach normal bilirubin levels, the role of PTBD was primarily to improve the quality of life of this group of patients. In our analysis, bilirubin level <120 µmol/L before drainage procedures was an independent predictive factor on overall survival, and a similar situation was seen with LDH <300 µmol/L, which was confirmed in multivariate analysis. It is known that high LDH levels are seen in metastatic cancer patients as well as in obstructive jaundice.

In our series, we observed successful biliary drainage with re-start of chemotherapy, and median survival time was significantly longer (from 9 to 64 weeks). However, these results are controversial. Similar studies reported a very wide interval of survival rate between 31 and 185 days [16,25,27,29]. In Ref. [20], patients with metastatic colorectal cancer had the greatest benefit from PTBD and additional chemotherapy, but in our study, they did not have a statistically significant longer survival rate. With regards to the survival rate of patients suffering from all types of biliary malignant obstructions collectively (pancreatic, gastric, cholangiocarcinoma and colon cancer), our results are comparable with the median survival rate of the studies of Vandabeele [20] and Van Leathems [30].

It is known that PTBD is an invasive procedure with related complications and mortality. In our series, we had 6.4% deaths within 7 days of procedure, and the 30-day mortality rate was 27.7%. In the literature, Vandenabeele [20], Garcarek [31] and Rees [32] reported 3%, 2.98% and 5.2% peri-interventional mortality (within 7 day), and for the 30-day mortality rate, Venderbeele [20], Rees [32] and Sut [33] reported 23.1%, 33%, and 43% (Table 5). 

Lauterio et al. [34] published the results of six studies in the management of PTBD in perihilar cholangiocarcinoma (h-CCA) with a mortality rate in the range of 0% [35] to 12% [36]. Among others, manipulation of the biliary tree has been commonly associated with cholangitis and septic complications [37], with a potential increase in post-operative morbidity and mortality [38].

All reported complications in our series caused prolonged hospitalization of patients with a negative impact on their quality of life.

Patient’s age (<60 years), type of primary tumor and numbers of previous lines of chemotherapy was without influence on the survival rate in our analysis.

The regression analysis in this series of patients identified significant predictive factors for longer survival after PBDT: good performance status (ECOG 0–1), additional chemotherapy, bilirubin level lower than 120 µmol/L, and LDH < 300 µmol/L.

In the published results of the meta-analysis that included three randomized trials and two observational studies that compared ERCP and EUS-BD for distal primary biliary decompression, where both techniques were found to be equally effective in achieving biliary drainage (ERCP) = 94.73% EUS = 93.67% and resolving jaundice (ERCP = 94.21% EUS = 91.23%, there were no significant differences in the overall rate of post-procedural-related adverse events (ERCP = 22.3% EUS = 15.2%, post procedural pancreatitis was significantly higher for ERCP (9.5% vs. EUS = 0%). There was no significant difference in the rates of reinterventions for jaundice between the groups (ERCP = 22.6% EUS = 15.2%) [39]. Other authors’ meta-analysis reported similar outcomes with no difference in reinterventions, procedure duration, stent patency and overall survival between the cohorts [40]. Similar to percutaneous methods, EUS is unlikely to be successful if the biliary ductal system is not dilated [6].

Almost all case series and randomized controlled trials examining the outcomes of PTBD in malignancy involved fewer than 100 patients.

The limitations of the presented results are related to the fact this was a retrospective analysis with heterogeneous primary tumors and different chemotherapy regimens.

## 5. Conclusions

PTBD is a safe and effective way to relieve jaundice caused by malignant tumors and should be considered in patients who are candidates for chemotherapy following drainage who could have prolonged patient survival. The best clinical outcome was seen in patients with good performance status and a lower level of bilirubin before drainage procedures. Improving the quality of life is a major goal in this palliative treatment but we need to assess the potential benefits and risk and select the best candidates for this procedure.

## Figures and Tables

**Figure 1 cancers-14-04673-f001:**
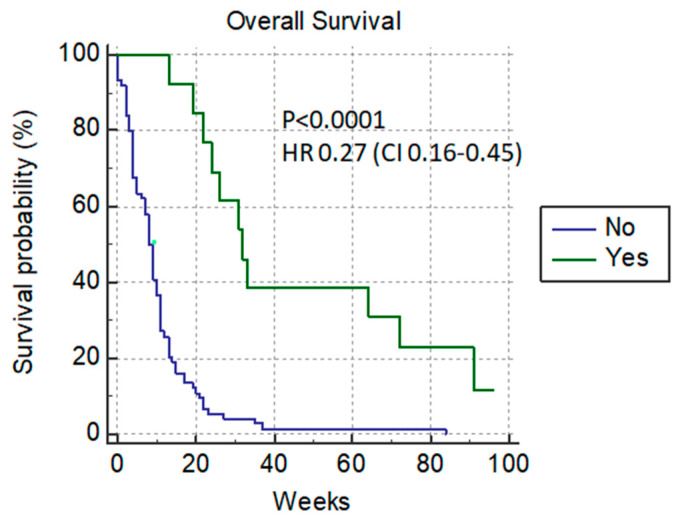
Normalized-bilirubin-level patients after PTBD.

**Figure 2 cancers-14-04673-f002:**
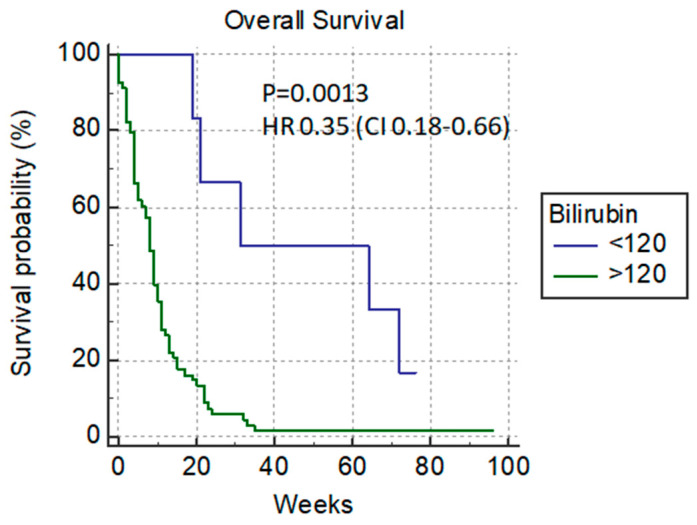
Overall survival versus bilirubin cut-off level of 120 µmol/L.

**Figure 3 cancers-14-04673-f003:**
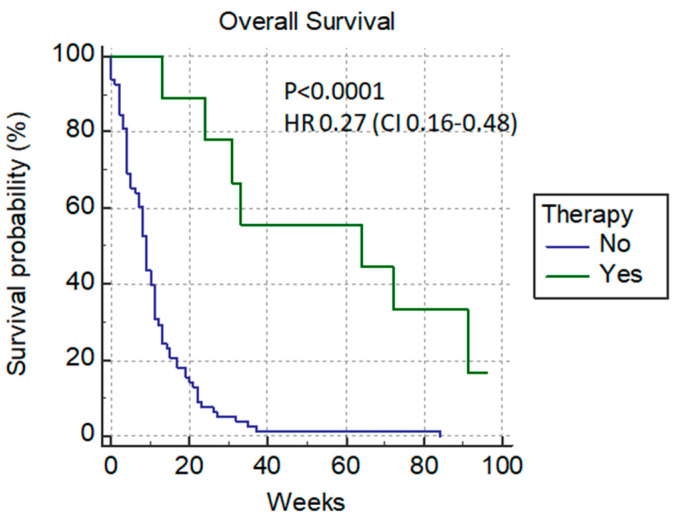
Additional chemotherapy after PTBD.

**Figure 4 cancers-14-04673-f004:**
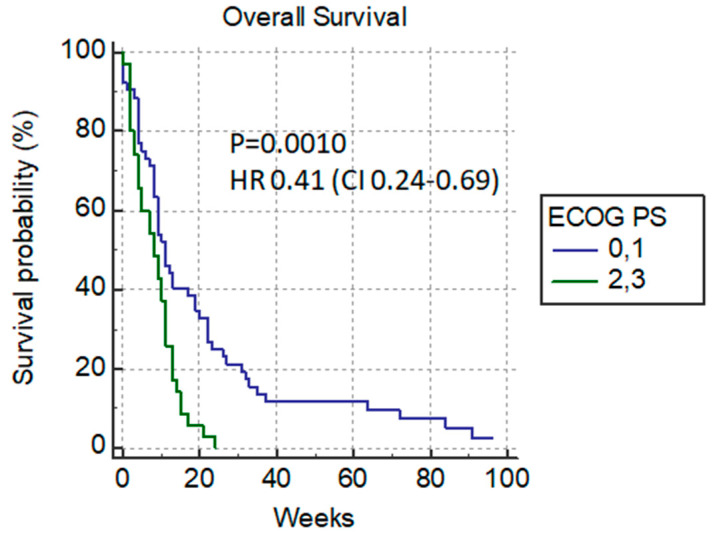
Overall survival versus ECOG status.

**Figure 5 cancers-14-04673-f005:**
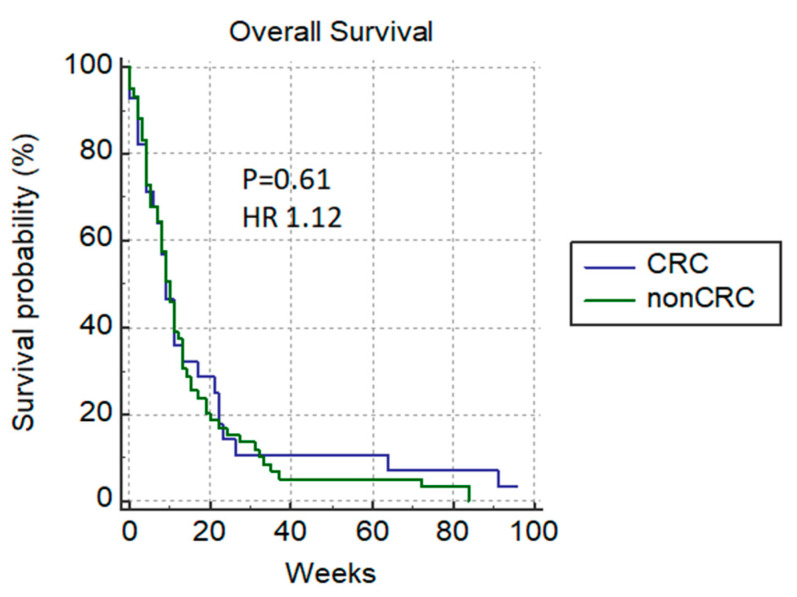
Overall survival of patients with colorectal versus non-colorectal cancer.

**Figure 6 cancers-14-04673-f006:**
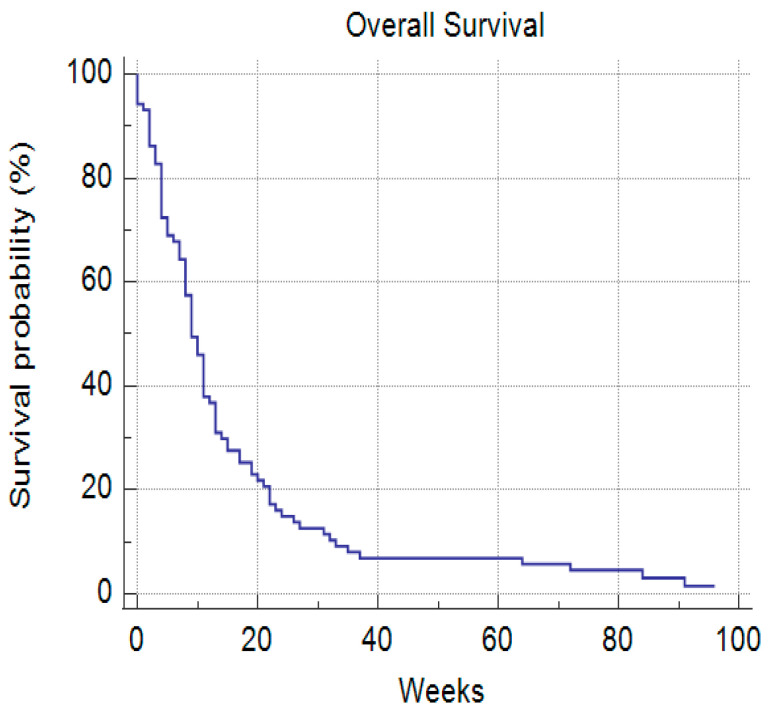
Median overall survival rate.

**Figure 7 cancers-14-04673-f007:**
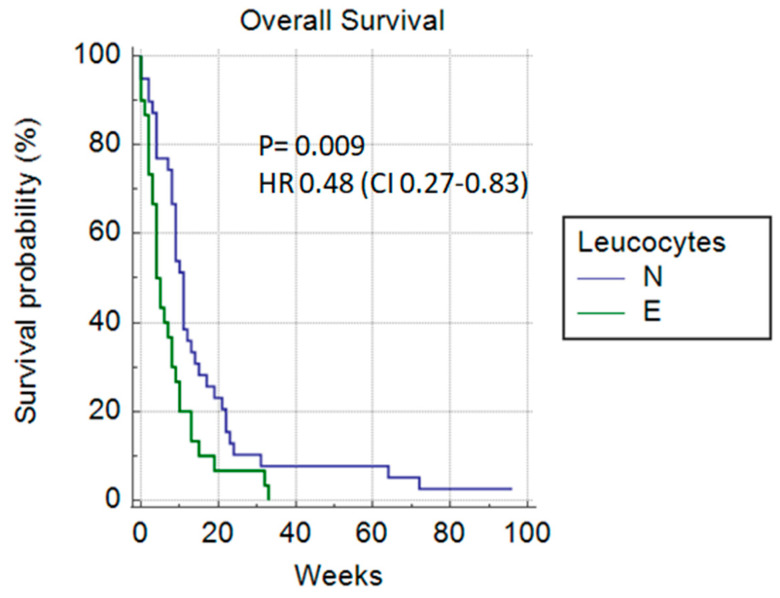
Leukocytes (normal vs. elevated).

**Figure 8 cancers-14-04673-f008:**
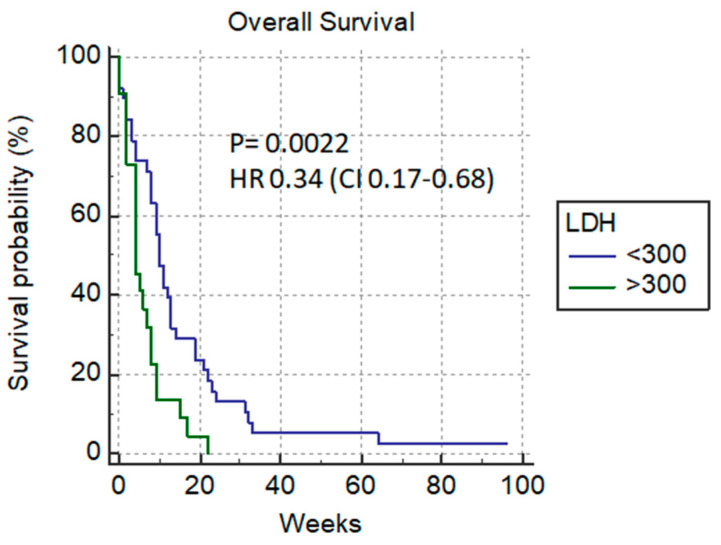
LDH level (<300 µmol/L vs. >300 µmol/L).

**Table 1 cancers-14-04673-t001:** Patients’ characteristics.

Variable		
**Patients**	Total number	89
	Gender male/female	55/34
	Mean age (time of PTBD) year	62.82
	Median level of bilirubin before PTBD	290 µmol/L
**Tumor**	Cholangiocarcinoma	28 (31.5%)
	Colorectal cancer	29 (32.6%)
	Pancreatic	22 (24.7%)
	Gastric	7 (7.8%)
	Other (breast, ovary, unknown primary)	3 (3.4%)
**Number of** **Procedures**	1	82
	2	6
	3	1
**ECOG performance status**	0	6
	1	46
	2	29
	3	8
	4	0

**Table 2 cancers-14-04673-t002:** Candidates for chemotherapy after PTBD.

	Candidates (ECOG 0–1) 52/89 (58.4%)	Not Candidates (ECOG 2–3) 37/89 (41.6%)
**Level of bilirubin ≤ 30 µmol/L**	12/52 (23%)	2/37 (5.4%)
**Received chemotherapy**	8/12	1/2
**Cholangiocarcinoma**	1	0
**Colorectal cancer**	6	0
**Pancreatic**	1	1
**Gastric**	0	0
**Did not receive chemotherapy**	4/12	1/2

**Table 3 cancers-14-04673-t003:** Multivariate analysis of LDH, leukocytes, bilirubin, ECOG and chemotherapy.

Covariate	b	SE	Wald	*p*	Exp(b)	95% Clof Exp(b)
**ABD**	−1.8059	0.7569	5.6925	0.0170 *	0.1643	0.0373 to 0.7244
**Therapy**	−0.9249	0.8374	1.2196	0.2694	0.3966	0.0768 to 2.0473
**ECOG group**	0.2216	0.2983	0.5522	0.4574	1.2481	0.6956 to 2.2394
**Leukocytes**	0.2050	0.3053	0.4507	0.5020	1.2275	0.6747 to 2.2331
**LDH groups**	0.4695	0.3243	2.0950	0.1478	1.5991	0.8468 to 3.0198
**Bilirubin**	0.4134	0.5881	0.4942	0.4821	1.5120	0.4775 to 4.7878

ABD—adequate bilirubin decline, Therapy—chemotherapy, LDH—lactate dehydrogenase, Leukocytes—white blood cells, ECOG—Eastern Cooperative Oncology Group. *, statistically significant difference.

**Table 4 cancers-14-04673-t004:** Complications after PTBD.

Complication	No.
**None**	65
**Bleeding**	5
**Pain**	12
**Cholangitis**	4
**Biliary leak**	9
**Death**	6
**Others (pancreatitis and abscess)**	2

**Table 5 cancers-14-04673-t005:** Clinical outcomes after PTBD.

Published Results	Additional Therapy	7-Day Mortality	30-Day Mortality
**Rees et al.** [32]	20–40%	5.2%	23.1%
**Vandenbeeele.** [20]	34%	3%	33%
**Garcareket al.** [31]	NA	2.98%	NA
**Sut et al.** [33]	NA	9.5%	43%
**Nikolic et al. [*]**	15.4%	6.4%	27.7%

* Present study.

## Data Availability

The datasets generated during and/or analyzed during the current study are available from the corresponding author on reasonable request.

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
