# Peer review of "The Clinical Benefit of Percutaneous Transhepatic Biliary Drainage for Malignant Biliary Tract Obstruction"

_cancers, 2022, doi:10.3390/cancers14194673_

Round 1
Reviewer 1 Report
The authors conduct a retrospective study regarding 89 patients receiving percutaneous transhepatic biliary drainage (PTBD) for biliary decompression of malignant biliary obstruction. The median survival time was 9 weeks. The patients received chemotherapy following PTBD had longer median survival time.
The authors concluded that patients with good performance status and lower bilirubin and LDH at the time of PTBD had the best clinical outcome and PTBD should be considered in patients who are candidates for chemotherapy 38 following drainage with prolonged life expectancy.
Comments:
Major:
1. As a retrospective intervention study, the most concerning point is the present study lack of comparison (control or other methods, such as medical only or endoscopy drainage. Thus, PTBD vs control group would be considered after a matched control group.
2. Malignant biliary tract obstruction could be caused by metastatic lymphadenopathy or liver mass. There should be a wide range of response to chemotherapy. Better chemotherapy would be predicted in colon cancer or pancreas cancer, and poor response to chemotherapy may be found in biliary cancer or gastric cancer. Did patients of all type of cancer benefit from PTGBD followed by chemotherapy?
3. The level of biliary tract obstruction and numbers of PTBD insertion should be clarified in the article. The distal biliary tract obstruction had higher clinical successful rate of drainage procedure. For tumor with obstruction level above common hepatic duct, such as cholangiocarcinoma bismuth type III and IV, more than one drainage are needed and the clinical outcome is often poor. The authors may clarify the type of biliary obstruction, distal or proximal?
4. The authors conclude several indicators for better benefit of PTBD, but this only provide information for better survival in patients with PTBD. As point 1, the authors need a control group for comparison to consolidate their conclusion.
5. The abstract should be revised according to the concepts of background, methods, results and conclusion.
Minor:
1.LDH< 300 level had significant longer survival rate. LDH is a non-specific maker of cell injury or tumor burden. More laboratory parameters or tumor characteristics (such as size, stage) should be included in the analysis. The authors should descript the results in detail and interpret the impacts of their findings to the clinical practice. Otherwise, the last paragraph of results (page 7, line 188-190): This section may be divided by subheadings. It should provide a concise and precise description of the experimental results, their interpretation, as well as the experimental conclusions that can be drawn. What is it means?
2.As the description in the article, the major goal of PTBD in palliative treatment is improvement of life of quality. The complications of PTBD, such bleeding, wound infection and impact of life of quality were not mentioned in the study.
3.In the second paragraph of Materials and Methods, why did the authors define the cut-off level of bilirubin at 120 μmol/l ?
4.Some mistakes should be corrected. We followed the time from disease onset to biliary__ obstruction (Page 2, line 82.). Obstructive jaundice caused by metastatic disease leads to many symptoms, a rapid deterioration of the performance status and a short survival. of advanced carcinoma. (page 8, line 195). Table 2: Cholangiocellulare means cholangiocarcinoma ? Not received chemotherapy 4/14 à 4/12 ?
5.the references should be updated.
Reviewer 2 Report
In this original paper entitles “The clinical benefit of the percutaneous transhepatic biliary drainage for malignant biliary tract obstruction” Nikolić and coll. provided a
Although the subject is interesting, the quality of the manuscript and of the stylistic presentation is still poor and needs major revisions:
Here are my concerns:
- Please, add the study endpoint at the end of the introduction section
- Please add the results of biliary draknage in cholangio (10.3390/cancers13153657) and pancreatic cancer (10.1111/den.14186), referring to the suggested papers
- Please, include the p values in the KM curves
- Fig 8 caption: the raph refers to LDH not LDL, please corrext
- Table 2: the patients who did not received chemo weee 4/12 not 4/14, please correct
- Lines 188-190 are residuals form the template!
- Multivariate model should include succesfull drainage and chemo
- Please complete style; author contributions etc
Best regards
Round 2
Reviewer 1 Report
the manuscript could not provide enough and effective information in patients with proximal malignant biliary obstruction.
Reviewer 2 Report
The authors properly revised the original manuscript
This revised version is suitable for publication